# Identification of Oxidative Stress-Related Biomarkers for Pain–Depression Comorbidity Based on Bioinformatics

**DOI:** 10.3390/ijms25158353

**Published:** 2024-07-30

**Authors:** Tianyun Zhang, Menglu Geng, Xiaoke Li, Yulin Gu, Wenjing Zhao, Qi Ning, Zijie Zhao, Lei Wang, Huaxing Zhang, Fan Zhang

**Affiliations:** 1Postdoctoral Research Station in Biology, Hebei Medical University, Shijiazhuang 050017, China; 2The Key Laboratory of Neural and Vascular Biology, Ministry of Education, Center for Brain Science and Disease, Hebei Medical University, Shijiazhuang 050017, China; 3Laboratory of Neurobiology, Hebei Medical University, Shijiazhuang 050017, China; 4Core Facilities and Centers, Hebei Medical University, Shijiazhuang 050017, China; 5Department of Biochemistry and Molecular Biology, College of Basic Medicine, Hebei Medica University, Shijiazhuang 050017, China

**Keywords:** oxidative stress, biomarker, pain, depression, comorbidity, bioinformatics

## Abstract

Oxidative stress has been identified as a major factor in the development and progression of pain and psychiatric disorders, but the underlying biomarkers and molecular signaling pathways remain unclear. This study aims to identify oxidative stress-related biomarkers and signaling pathways in pain–depression comorbidity. Integrated bioinformatics analyses were applied to identify key genes by comparing pain–depression comorbidity-related genes and oxidative stress-related genes. A total of 580 differentially expressed genes and 35 differentially expressed oxidative stress-related genes (DEOSGs) were identified. By using a weighted gene co-expression network analysis and a protein–protein interaction network, 43 key genes and 5 hub genes were screened out, respectively. DEOSGs were enriched in biological processes and signaling pathways related to oxidative stress and inflammation. The five hub genes, RNF24, MGAM, FOS, and TKT, were deemed potential diagnostic and prognostic markers for patients with pain–depression comorbidity. These genes may serve as valuable targets for further research and may aid in the development of early diagnosis, prevention strategies, and pharmacotherapy tools for this particular patient population.

## 1. Introduction

Psychiatric patients with comorbid pain disorders not only face challenges within the health care system, but also experience complex clinical manifestations. These patients often suffer from a combination of psychiatric disorders such as depression, anxiety, post-traumatic stress disorder (PTSD), and schizophrenia, which coexist with conditions like neuropathic pain, fibromyalgia, and somatic symptom disorder. The relationship between chronic pain and depression is intricate and bidirectional. On the one hand, depression can be a risk factor for the development of chronic pain [1]. On the other hand, chronic pain can also contribute to the development of depression [2,3]. When patients experience chronic pain, it can exacerbate their psychiatric symptoms, causing increased distress, impairment in daily functioning, and an overall decrease in quality of life [4]. Conversely, undertreated psychiatric conditions can intensify the perception of pain and impede the individual’s ability to cope with it, hindering their rehabilitation efforts. Given the complexity of these cases, it is crucial to have comprehensive assessment and management strategies in place. The comorbidity of chronic pain and depression has long been recognized in the clinical setting [5,6]. Additionally, preclinical studies of chronic pain have reported depression-like behaviors [7,8], indicating a shared neural basis for chronic pain and depression.

Oxidative stress refers to an imbalance between production of reactive oxygen species (ROS) and antioxidant defense mechanisms of the body. It has been identified as a significant factor in the development and progression of psychiatric and pain disorders. In various psychiatric disorders such as depression, anxiety, and schizophrenia, markers of oxidative stress are often found to be elevated, indicating a potential role in the manifestation and worsening of these conditions. Both clinical and preclinical studies have provided evidence supporting the close relationship between major depressive disorder (MDD) and an imbalance characterized by increased oxidative stress and decreased antioxidant defenses [9,10,11]. The pathophysiology of both unipolar and bipolar depression also involves oxidative and nitrosative stress, in addition to immune-inflammatory mechanisms [12]. Similarly, chronic pain disorders such as neuropathic pain and fibromyalgia have been linked to increased oxidative stress, which may contribute to the intensification of pain perception and the perpetuation of pain pathways. Multiple studies have shown that various animal models of chronic pain exhibit indications of oxidative stress and mitochondrial dysfunction [13,14]. In the case of chronic migraine and neuropathic pain associated with central sensitization, altered levels of plasma oxidative stress biomarkers have been observed [15]. It is important to note that the relationship among oxidative stress, psychiatric disorders, and pain disorders is complex and bidirectional.

Bioinformatics tools and algorithms have played a pivotal role in decoding and analyzing the vast amount of genomic data generated from next-generation sequencing technologies. This has led to high-throughput quantification of genetic alterations and has opened new avenues for precision medicine. Moreover, bioinformatics techniques are instrumental in analyzing complex biological networks and pathways, shedding light on the underlying molecular mechanisms of diseases. Bioinformatics analyses have been utilized in animal models and patients and have uncovered several biomarkers and signaling pathways in correlation with pain–depression comorbidity [16,17,18]. However, the underlying mechanisms that contribute to pain–depression comorbidity are not yet fully understood, especially the pathways related to oxidative stress. In order to improve the reliability and repeatability of this research, we first compared pain–depression comorbidity-related genes and oxidative stress-related genes from two separate datasets. Subsequently, the differentially expressed genes (DEGs) and differentially expressed oxidative stress-related genes (DEOSGs) were used as the inputs for the weighted gene co-expression network analysis (WGCNA) and protein–protein interaction (PPI) approaches to identify key genes and hub genes, respectively (Figure 1). The functional enrichment analysis suggested that these genes could serve as potential indicators of oxidative stress and inflammation levels in predicting the diagnosis of both disorders simultaneously. Understanding the oxidative stress-related pathways underlying the relationship between these two disorders will contribute to the development of effective diagnostic and therapeutic strategies for individuals suffering from both pain and depression.

## 2. Results

### 2.1. Identification of DEGs and DEOSGs

Based on analysis of the provided data, a total of 580 genes were identified as being differentially expressed between the Low Pain and High Pain groups. Out of these genes, 178 genes were found to be up-regulated, meaning they were expressed at higher levels in the High Pain group compared with the Low Pain group. The remaining 402 genes were down-regulated, indicating lower expression levels in the High Pain group. To further investigate the potential relationship between pain and oxidative stress, an intersection analysis was conducted between the 580 DEGs and a set of 1216 oxidative stress-related genes (OSGs, Figure 2A,B). This analysis aimed to identify genes that were both differentially expressed and related to oxidative stress. As a result, 35 genes were found to be DEOSGs—CAT, TXN, GTPBP3, FOXO1, FOS, ERN1, MAP2K4, RPS6KA5, MAP2K1, MYC, ALOX5, THBD, EGR1, TNFRSF1A, AOC3, SGK1, CAMK2G, S100A8, DMD, NDRG1, RAF1, IGF1R, MFN2, HBEGF, NUDT5, CCL5, GSTZ1, ATXN2, SREBF1, CXCR4, BACH2, SLPI, PLCG2, DNAJC3, and SURF1 (Figure 2C). This finding suggests a possible link between oxidative stress and pain–depression comorbidity, as the identified DEOSG may contribute to the development or regulation of diseases in the High Pain group.

### 2.2. Enrichment Analysis of KEGG Pathways and GO Functions for DEOSGs

To gain a more comprehensive understanding of the biological functions of the identified DEOSGs, we performed KEGG and Gene ontology (GO) enrichment analyses. We selected the top ten KEGG terms and five significantly enriched GO terms. The results of the enrichment analysis revealed that the top ten significant KEGG pathways associated with the DEOSGs included MAPK, ErbB, GnRH, TNF, FoxO, the Fc epsilon RI signaling pathway, Kaposi sarcoma-associated herpesvirus infection, proteoglycans in cancer, bladder cancer, and glioma (Figure 3A). The enrichment analysis findings also indicated that the DEOSGs were involved in various biological processes including peptidyl-tyrosine phosphorylation, peptidyl-tyrosine modification, response to mechanical stimulus, response to carbohydrate, and peptide secretion (Figure 3B). Moreover, cellular component analysis showed that DEOSG were mainly related to the secretory granule lumen, cytoplasmic vesicle lumen, and vesicle lumen (Figure 3B). Additionally, the analysis of molecular function highlighted the significant enrichment of DEOSGs in protein serine kinase activity, protein serine/threonine kinase activity, protein kinase regulator activity, kinase regulator activity protein, and serine/threonine/tyrosine kinase activity (Figure 3B).

### 2.3. Screening of Pathway Enrichments by GSEA

Gene set-enrichment analysis (GSEA) was employed as a tool to determine the enrichment of gene sets and pathways associated with a group of up-regulated or down-regulated genes in the High Pain group. The analysis of enriched entries between the Low Pain and High Pain groups in gene set GSE127208 showed a total of 44 KEGG pathways and 707 GO biological processes (GOBPs) that were significantly associated with pain. Specifically, up-regulated genes were found to be enriched in several KEGG pathways including ECM–receptor interaction, gap junction, neuroactive ligand–receptor interaction, and olfactory transduction (Figure 4A). These pathways are known to play crucial roles in sensory perception, chemical stimulus detection, and neurotransmission. In terms of GOBPs, the up-regulated genes showed significant enrichment in processes related to sensory perception of chemical stimulus, sensory perception of smell, sodium ion homeostasis, sodium ion transmembrane transport and vascular process in circulatory system (Figure 4B). These findings suggest that the up-regulated genes may be involved in the regulation of sensory perception and sodium ion levels, which are closely related to pain sensation and vascular function. On the other hand, the down-regulated genes were observed to be extensively enriched in KEGG pathways such as epithelial cell signaling in helicobacter pylori infection, Fc gamma R-mediated phagocytosis, oxidative phosphorylation, ribosome, and spliceosome (Figure 4C). These pathways are known to be involved in various cellular and molecular processes such as cell signaling, immune response, energy production, protein synthesis, and RNA processing. In terms of GOBPs, the down-regulated genes were significantly enriched in processes including B-cell activation, cytoplasmic translation, ribonucleoprotein complex biogenesis, ribonucleoprotein complex subunit organization and RNA splicing via transesterification reaction (Figure 4D). These processes are important for the proper functioning and regulation of proteins and RNA molecules in the cells. Overall, the results of GSEA provide an in-depth understanding of molecular pathways and biological processes that may be involved in the differences in pain perception between the Low Pain and High Pain groups in the studied gene set.

### 2.4. Construction of the WGCNA Network and Identification of Key Genes

In order to identify the genes highly correlated with pain and oxidative stress, a WGCNA was conducted. We assessed the scale independence and average connectivity and found that β = 7 was the appropriate soft threshold (Figure 5A). Using this power, we generated four gene co-expression modules (Figure 5B). In particular, the MEyellow was chosen (correlation coefficient > 0.2, *p* < 0.05, Figure 5C) [19]. We identified a total of 223 genes within this module that showed potential as biomarkers for pain-related oxidative stress. To further explore the importance of these genes, we performed a Venn diagram analysis to determine the relationship between the 223 genes and the 580 DEGs from the GSE127208 dataset. This analysis revealed 43 overlapping DEGs that were considered as key genes (Appendix A). Notably, two genes in these key genes (DNAJC3 and FOS) were also identified as DEOSGs (Figure 5D). As a result, we selected these 43 key genes for subsequent analyses.

### 2.5. Enrichment Analysis of Key Genes

We performed GO and KEGG enrichment analyses on 43 key genes using the “clusterProfiler” R package. As a result, we identified a total of 144 GOBPs and 5 KEGG pathways that were significantly enriched. To further investigate these enrichments, we focused on the top ten GOBPs and KEGG pathways based on their *p*-values (Figure 6). The analysis result revealed that the key genes were predominantly involved in various biological processes, including regulation of GTPase activity, which plays a crucial role in controlling cellular signaling and intracellular transport. Additionally, the small GTPase-mediated signal transduction pathway that regulates multiple cellular processes, was also enriched among the key genes. Another significant biological process enriched in the key genes was adenylate cyclase-activating G protein-coupled receptor signaling pathway. This pathway is responsible for transmitting extracellular signals into cells and plays a critical role in various physiological processes. Moreover, the analysis revealed an enrichment of genes involved in glucan catabolic process, cellular polysaccharide catabolic process, polysaccharide catabolic process, and cellular carbohydrate catabolic process. These processes are responsible for breakdown of complex carbohydrates and subsequent utilization of the released sugars by cells. The key genes were also enriched in the process of positive regulation of substrate adhesion-dependent cell spreading. This process plays a crucial role in cell migration, tissue development, and wound healing by promoting attachment and spreading of cells on different substrates. (Figure 6A). There are five significant KEGG pathways of key genes, including metabolism of starch and sucrose, which is fundamental for carbohydrate utilization; the parathyroid hormone synthesis, secretion, and action pathway, which regulates calcium and phosphate homeostasis; the vascular smooth muscle contraction pathway, highlighting their potential involvement in the regulation of vascular tone and blood pressure; the estrogen signaling pathway, which plays a critical role in various physiological processes, including reproductive development and maintenance; and the Yersinia infection pathway, indicating that the key genes may be associated with the host response to Yersinia bacterial infections (Figure 6B).

### 2.6. Construction of the PPI Network and Screening of Hub Genes

In order to explore the interaction between the proteins encoded by key genes, we constructed the PPI network consisting of 38 nodes and 78 edges (Figure 7A). The CytoHubba plugin was utilized to identify the most important genes in this network, which prioritized the top 10 genes based on their significance in the PPI network. Four different topological analysis methods were employed in order to further narrow down the hub genes. These methods included maximum clique centrality (MCC), maximum neighborhood component (MNC), edge percolated component (EPC), and Degree. The results pointed towards five specific hub genes. These hub genes, namely RNF24, MGAM, FOS, TKT, and SIGLEC5, were determined to be the most influential and central genes in the PPI network (Figure 7A,B).

### 2.7. Correlation Analysis between Hub Genes and Immune Cells

Based on KEGG pathways analysis, it was found that the DEGs were highly enriched in pathways related to immune response (as depicted in Figure 3 and Figure 4). We employed the CIBERSORT algorithm to estimate the relative abundance of 22 immune cell types in each sample and to further investigate the role of immune cells in the context of pain. By comparing the proportions of these immune cell types, we identified significant differences in abundance of three particular types of immune cells namely CD8 T cells, naive CD4 T cells, and NK cells resting, between the Low Pain and High Pain groups (Figure 8A). Spearman correlation analysis was conducted to explore the potential relationship between the five hub genes and infiltration of immune cells. The results revealed that the MGAM gene exhibited a negative correlation with all three immune cell types, and the FOS gene displayed a significantly negative correlation with CD8 T cells and naive CD4 T cells, indicating that their expression levels were inversely associated with the abundance of these immune cells (Figure 8B).

### 2.8. Validation of Hub Gene Expression

In addition to the datasets of depression patients, we selected datasets of other psychiatric patients with bipolar disorder, schizophrenia, schizoaffective disorder, combined schizophrenia and schizoaffective disorder, or PTSD to further validate the expression levels of these hub genes. The findings revealed that, in the dataset GSE127208, all the hub genes, except for SIGLEC5, exhibited a significant decrease in expression levels in the High Pain group compared with the Low Pain group (Figure 9).

### 2.9. Identification of Transcription Factors, microRNAs and Drugs Regulating Key Genes

We conducted further analysis to explore the regulatory relationships between hub genes and other small molecules, specifically focusing on the regulatory network connections with transcription factors (TFs). In order to assess the effects of microRNAs (miRNAs) and TFs on the transcriptional levels of hub genes, we generated miRNA and TF interaction networks with the identified hub genes and visualized them using Cytoscape software. Our findings revealed that a total of 18 miRNAs and 2 miRNAs were found to interact with RNF24 and FOS, respectively (Figure 10A). Additionally, we identified three TFs that interacted with RNF24, two TFs with MGAM, FOS, and TKT, and one TF with SIGLEC5 (Figure 10B). Interestingly, our analysis revealed that three TFs, namely POU2F1, NFIA, and SREBF2, had interactions with two hub genes, while PITX2 was found to interact with three hub genes. These findings suggest that these specific TFs may potentially have a closer regulatory relationship with the hub genes.

We utilized the drug–gene network from the drug–gene interaction database (DGIdb) to identify potentially effective drugs that could target the key genes. By visualizing this network using Cytoscape software, we are able to gain a comprehensive understanding of the relationship between genes and drugs. In our analysis, we discovered that two hub genes, FOS and MGAM, exhibited intersections with a significant number of drugs and chemicals (Figure 10C). Specifically, FOS had connections with 10 different drugs or chemicals, and MGAM had intersections with 10 drugs or chemicals as well. This finding suggests that FOS and MGAM may be key players in the effectiveness of these drugs, making them potential targets for therapeutic intervention.

## 3. Discussion

The etiology and pathophysiology of pain–depression comorbidity are heterogeneous and intricate. Numerous detailed studies have established the connection between pain and psychiatric diseases such as anxiety, depression, schizophrenia, and bipolar disorder, highlighting oxidative stress as a common link [20,21]. However, it poses challenges in accurately determining the oxidative stress status as a reliable biomarker in pain–depression comorbidity. Thus, it is crucial to develop a quantitative assessment method using specific biomarkers that can reflect the patients’ condition and the effectiveness of treatments. In this study, we specifically aimed to investigate the oxidative stress-related DEGs of MDD patients in the GSE127208 dataset, which consists of MDD and other psychiatric disorders [18]. By analyzing the intersection between the 580 DEGs and 1216 OSGs, we identified 35 DEOSGs. To further pinpoint specific genes of interest, we utilized WGCNA, the PPI network, and topology analysis. As a result, we discovered 43 key genes associated with pain–depression comorbidity, along with the identification of RNF24, MGAM, FOS, TKT, and SIGLEC5 as hub genes. Overall, by unraveling the molecular basis and key genes involved in this complex comorbidity, this study paves the way for further research and potentially more therapeutic interventions targeting oxidative stress.

The result of KEGG pathway enrichment for DEOSGs showed that the FOS gene was involved in both MAPK and TNF signaling pathways that were among the top 10 KEGG pathways identified. The activation of the MAPK pathway in nociceptive neurons has been found to contribute to the generation of pain hypersensitivity in both transcription-dependent and transcription-independent manners. Therefore, targeting the MAPK signaling pathways in primary afferents and spinal cord could be a promising strategy for the development of novel analgesics. Studies have shown that activation of FOS and MAPK occurs in response to pain stimuli in various contexts [22,23]. The activation of MAPK leads to increased phosphorylation and nuclear translocation of TFs such as JUN and FOS [24] and is implicated in the development and maintenance of neuropathic pain [25]. Additionally, TNFα and MAPK have been identified as central regulators of persistent functional pain and neuroinflammation [26]. Inhibition of either TNFα or MAPK is sufficient to reduce mechanical allodynia [27], suggesting TNFα and MAPK play a sequential role in the process of inducing neuropathic pain. Regarding the top enriched GOBPs, the FOS gene appears to be particularly relevant to the response to mechanical stimulus, which suggests that the FOS gene may play a role in the body’s reaction to physical forces or pressure. We also recruited another tool called GSEA to detect significantly enriched pain-related KEGG pathways and GOBPs. The gap junction, composed of connexins, is a critical component in neuronal plasticity that contributes to the development of pain hypersensitivity [28,29]. A recent study has revealed that the gap junction and TNF-α exert their biological effects on neuropathic pain through distinct mechanisms of action [30]. Furthermore, mitochondrial dysfunction has been proposed as a potential causal or contributory mechanism in chronic pain. Mitochondrial oxidative phosphorylation is a process that involves transfer of electrons through mitochondrial electron transport chain (mETC) and uptake of molecular oxygen, ultimately leading to the generation of ATP. However, it is important to note that during this process, electrons can leak from the mETC and react with molecular oxygen, resulting in the production of superoxide radicals that are a type of ROS. ROS and oxidative stress, may contribute to the development of neuropathic and inflammatory pain [31]. Moreover, mediators released from injured tissues can transiently activate G protein-coupled receptors (GPCRs) located at the plasma membrane of neurons, leading to acute hyperexcitability and nociception. Activation of GPCRs can also induce immune cells to release pro-inflammatory factors, cause damage to nerve cells, and exacerbate pain [32]. Based on the KEGG enrichment analysis of key genes, it has been found that several hub signaling pathways, namely, parathyroid hormone synthesis, secretion and action, estrogen signaling, and Yersinia infection, are closely related to pain and may play a significant role in pain management [33,34,35,36,37]. To summarize, these findings collectively suggest that oxidative stress and inflammation likely play a crucial role in the development and persistence of pain. Additional research is necessary to comprehensively comprehend the mechanisms and functions of these factors in the perception and treatment of pain.

In this study, we identified five hub genes that play important roles in pain–depression comorbidity. Specifically, RNF24 interacts with transient receptor potential cation channel subfamily C (TRPC) proteins in the Golgi apparatus, influencing their intracellular trafficking [38]. Dysfunction of TRPC channels has been linked to psychiatric disorders and inflammatory pain [39,40]. Blocking TRPC channels has been shown to alleviate nociception, inflammatory pain hypersensitivity [41], and anxiety-like behavior associated with nerve injury [42]. Additionally, levels of FOS protein increased in patients with neuropathic pain related to orofacial cancer [43]. TKT encodes a thiamine-dependent enzyme involved in the pentose phosphate pathway, which plays a role in glucose metabolism. Thiamine deficiency affects the function of thiamine pyrophosphate-dependent enzymes, leading to decreased ATP production, increased oxidative stress, and neurological deficits. These findings provide valuable insights into the role of oxidative stress and mitochondrial dysfunction in the development of neurodegenerative disorders [44]. MGAM encodes maltase-glucoamylase, an enzyme responsible for digesting starch and regulating gluconeogenesis [45]. SIGLEC5 inhibits the activation of monocytes, macrophages, and neutrophils. We found that these hub genes are down-regulated in patients with pain–depression comorbidity, indicating that they may act as pain suppressor genes. This was further validated in other datasets of psychiatric patients with high pain severity scores where similar down-regulation of the hub genes was observed, except for SIGLEC5. These genes provide insights into the biological functions of oxidative stress in pain–depression comorbidity and offer potential diagnostic and prognostic tools for managing patients with comorbid pain and psychiatric disorders. However, further biological experiments are necessary to confirm the molecular mechanisms of the hub genes and the relationship between oxidative stress biomarkers and the co-occurrence of pain–depression comorbidity in clinical practice.

It is hypothesized that oxidative stress plays a crucial role in intensifying pain by exacerbating the typical pathological reactions, such as inflammation and neuropathy. These reactions, when aggravated by oxidative stress, further contribute to the development and persistence of pain. When our body experiences chronic inflammation, it triggers a cascade of biological processes that result in the manifestation of various symptoms including chronic pain that is believed to be caused by the release and accumulation of ROS during the immune response. ROS are produced by immune cells when they consume increased amounts of oxygen in order to carry out their immune functions. This excessive production of ROS can lead to oxidative stress and damage to the surrounding tissues and cells, thus triggering and sustaining chronic pain [46]. This leads to a cycle where pro-inflammatory mediators are produced, activating and recruiting additional inflammatory immune cells, which then generate more ROS [47]. Multiple factors have been identified to influence oxidative stress-induced inflammation, including TFs, chemokines, inflammatory cytokines, and microRNAs. These compounds not only promote the release of reactive species, but they also stimulate nociceptors and afferent pain neurons, further contributing to chronic pain [20]. However, recent evidence reveals that certain subtypes of immune cells can actually reduce the pain experience [48]. Specifically, CD8^+^ T cells, CD4^+^ T cells, and NK cells are found to be required for the resolution of neuropathic pain [49,50,51]. Consistently, immune cells like CD8^+^ T cells, naive CD4^+^ T cells, and resting NK cells were found to have negative correlation with certain hub genes in our study, suggesting that immune cells and oxidative stress-related genes may play distinct roles in pain. Therefore, additional research is required to comprehensively investigate and ascertain the precise nature and significance of immune cells’ involvement in conjunction with oxidative stress in the development and persistence of chronic pain.

Furthermore, we constructed a gene regulatory network to gain a better understanding of the molecular mechanisms behind the regulation of oxidative stress-related hub genes. This network allowed us to identify key TFs that are responsible for controlling gene transcription by recognizing specific DNA sequences. In addition to TFs, we also investigated the role of miRNAs in controlling gene expression. miRNAs accomplish this by binding to specific sites on mRNA transcription sequences. By doing so, they inhibit the translation of mRNA or regulate the activity of TFs, thus influence various cellular and biological processes. We found a total of 20 miRNAs and 10 TFs in the miRNA–gene and TF–gene interaction networks, indicating their potential involvement in the regulation of oxidative stress-related hub genes. To further explore potential therapeutic options, we identified 20 drugs that have the potential to target these key genes. Among these drugs, antibiotic [52], paclitaxel [53], daidzein [54], nimodipine [55], baclofen [56], alcohol [57], and baicalein [58] have already been proven to be associated with oxidative stress in pain. Therefore, these drugs have the potential in the field of molecular targeted therapy, specifically designed to alleviate pain conditions related to oxidative stress.

There are some concerns and limitations that deserve to be mentioned. Firstly, the results of this study are limited by the datasets and analyzing methods used. It is necessary to utilize alternative methods and replicate the analysis results in different datasets. Secondly, this study failed to consider the dynamic changes in gene expression levels, especially in the case of pain–depression comorbidity. Long-term disease progression or treatment may lead to dynamic changes in gene expression, which needs to be fully considered in future studies. Last but not the least, although this study proposes some potential biomarkers and molecular signaling pathways, these results still need to be validated in clinical samples. Direct verification of oxidative stress biomarkers through quantitative real-time PCR experiment in clinical samples will greatly improve the clinical applicability and reliability of the data.

## 4. Materials and Methods

### 4.1. Data Collection

The dataset (GSE127208) was downloaded from the Gene Expression Omnibus (GEO) database (https://www.ncbi.nlm.nih.gov/geo/ (accessed on 19 January 2024)). From a total of 674 samples, 385 samples were selected according to pain scale score, and divided into High Pain (pain scale 6–10) and Low Pain (pain scale 0–2) groups [18]. Seventy-eight MDD samples (High Pain 42, Low Pain 36) were selected as the analysis set. Another 306 (one sample with no clear disease description was not included) samples were used for the validation set. The demographic characteristics of the studied subjects are listed in Appendix A. According to the GPL570 [HG-U133_Plus_2] Affymetrix Human Genome U133 Plus 2.0 Array chip platform, the downloaded corresponding mRNA probe identification numbers from the dataset (GSE127208) were mapped to corresponding official gene symbols. Probe sets that did not have a corresponding gene symbol were eliminated. If multiple probes were annotated with the same symbol, the mean value was adopted. The transformed gene expression matrix was used for further analysis. The OSGs were obtained from the GeneCards database with a relevance score ≥ 7.

### 4.2. Screening of DEGs

Differentially expressed genes between High Pain and Low Pain groups in MDD samples were obtained using the “limma” R package (version 3.48.3), and were selected according to *p*-value cutoff of 0.05 and no fold change cutoff because of the subtle expression differences observed [59]. Differentially expressed oxidative stress-related genes were obtained through Venn plots analysis using the R VennDetail package (version 1.2.0).

### 4.3. Functional Enrichment Analysis for DEOSGs

GO enrichment analysis and Kyoto Encyclopedia of Genes and Genomes (KEGG) enrichment analysis were performed to reveal the biological functions and pathways related to DEOSGs using the “clusterProfiler” R package (version: 3.16.0).

### 4.4. Gene Set-Enrichment Analysis

To determine the significantly enriched entries in the High Pain groups, GSEA was conducted using the “clusterProfiler” R package (version 4.4.4) that is a powerful tool widely used for functional enrichment analysis in genomics research. The GSEA utilized the KEGG and GOBP datasets obtained from the MSigDB database as background datasets. The KEGG dataset contains curated pathway information, while the GOBP dataset contains gene ontology terms associated with biological processes.

### 4.5. Weighted Gene Co-Expression Network Analysis

Weighted gene co-expression network analysis, a system biology strategy adopted for identifying co-expressed gene modules and exploring their correlation with disease phenotypes, was performed to screen for module genes with pain traits. Feature genes with a coefficient of variation > 0.05 were selected for WGCNA. The “WGCNA” R package (version 1.71) was applied to identify and select the key modules with the highest correlation with pain phenotypic traits.

### 4.6. Identification of Key Genes

To identify key genes, we applied the R VennDetail package (version 1.2.0) to generate Venn diagrams that allowed us to visually determine the intersection of genes between the most pain-relevant key module identified through WGCNA and the DEGs or DEOSGs. The intersecting genes found in this analysis were considered as key genes, as they demonstrated a significant association with pain.

### 4.7. Functional Enrichment Analysis for Key Genes

GO enrichment analysis and KEGG enrichment analysis were performed to reveal the biological functions related to key genes using the “clusterProfiler” R package (version: 3.16.0).

### 4.8. PPI Network Construction and Topology Network Interaction Analysis

The STRING (version: 12) database was used to predict and analyze whether there is an interaction between the proteins encoded by key genes. The PPI score was set to 0.15. The constructed PPI network was imported into the Cytoscape software (version: 3.6.1) platform to be analyzed and visualized. Four topological analysis algorithms MCC, MNC, Degree, and EPC in the CytoHubba plugin were used to predict and explore top 10 genes in the PPI network, and the intersected genes of the top 10 genes were considered to be hub genes.

### 4.9. Immune Cell Infiltration Analysis

Relative infiltration abundance of 22 types of immune cells in each sample was obtained using the CIBERSORT algorithm. The spearman correlation coefficient between expression level of hug genes and immune cells with significantly differential distribution was calculated using cor function in R language.

### 4.10. Exploration of miRNAs and TFs Regulating Hub Genes

Hub gene-related miRNA-target regulation edges were predicted and listed using the miRWalk 3.0 database. The regulation edges were used to construct the regulation network, which appeared in the miRDB and miRTarBase databases and the score was more than 0.95. Hub gene-related TFs -mRNA edges were predicted using TRANSFAC and JASPAR in Enrichr software (version: 3.2). Furthermore, the TF–gene and miRNA–gene interaction networks were visualized by Cytoscape.

### 4.11. Identification of Potentially Effective Drugs Targeting Hub Genes

Potentially effective drugs targeting key genes were obtained from the DGIdb and visualized by Cytoscape.

### 4.12. Statistical Analysis

All statistical analyses were performed using the R programming language. Independent-samples *t*-test or the Kruskal test was used to compare data between different groups. *p* < 0.05 was considered statistically significant.

## 5. Conclusions

In summary, through comprehensive bioinformatics methods, we identified five hub genes, all of which are associated with oxidative stress. Moreover, we constructed a regulatory network to understand the interactions between these hub genes and explored potential drugs that target these hub genes. These hub genes could serve as oxidative stress biomarkers in predicting diagnosis of pain–depression comorbidity. It is important to note that while our study provides significant insights, further validation is required through more prospective studies, alternative analysis methods and additional experiments. The inclusion of longitudinal studies and clinical trials will be critical to confirm both the diagnostic and prognostic value of these hub genes and to determine the efficacy of targeted therapies in clinical settings.

## Figures and Tables

**Figure 1 ijms-25-08353-f001:**
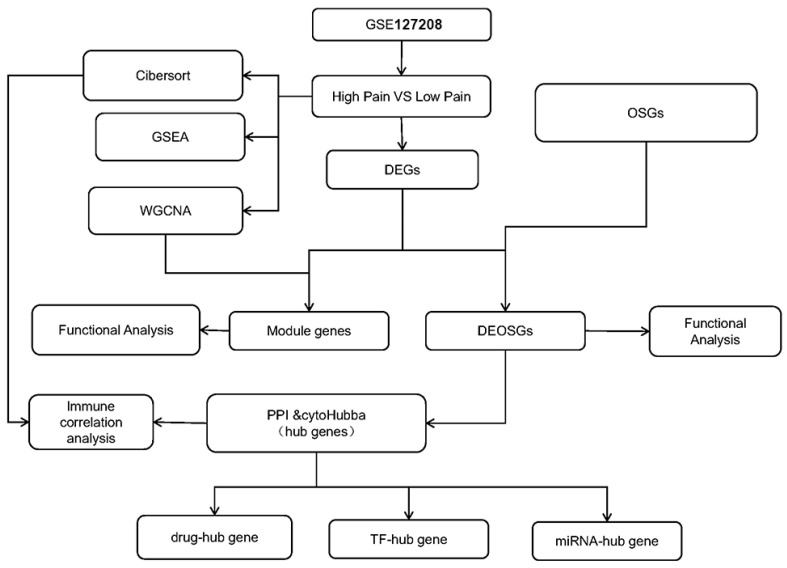
Flowchart of this study. OSGs, oxidative stress-related genes; DEGs, differentially expressed genes; DEOSGs, differentially expressed oxidative stress-related genes; GSEA, gene set-enrichment analysis, WGCNA, weighted gene co-expression network analysis, PPI, protein–protein interaction network; miRNA, microRNA; TF, transcription factor.

**Figure 2 ijms-25-08353-f002:**
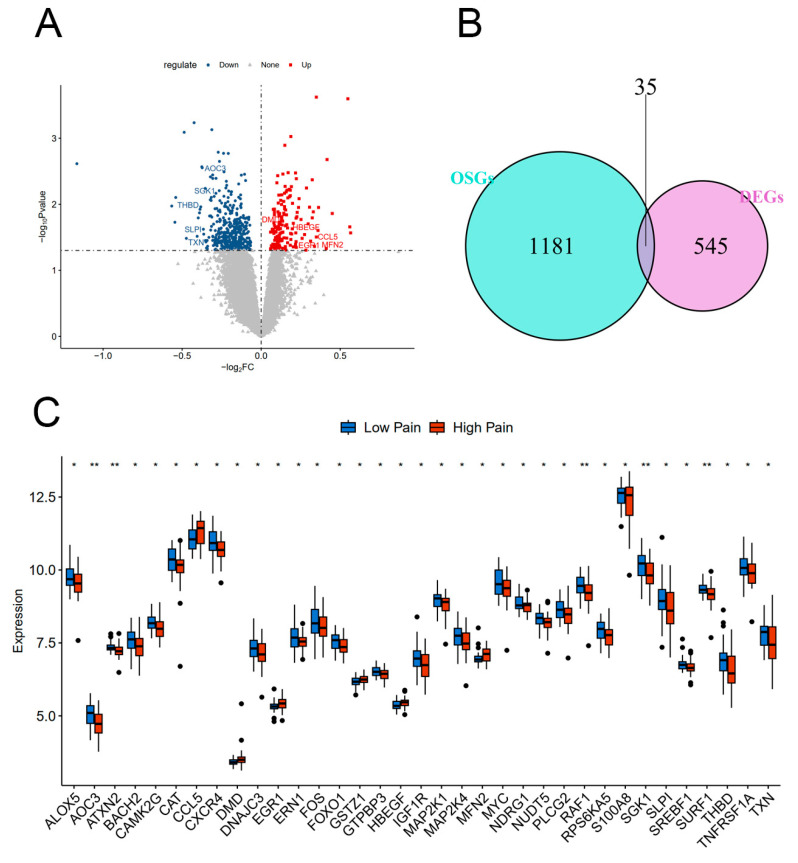
DEGs and DEOSGs between the Low Pain and High Pain groups. (**A**) Volcano plot displaying DEGs. (**B**) Venn diagram displaying DEOSGs. (**C**) Gene expression level of DEOSGs between the Low Pain and High Pain groups from the GSE127208 dataset.

**Figure 3 ijms-25-08353-f003:**
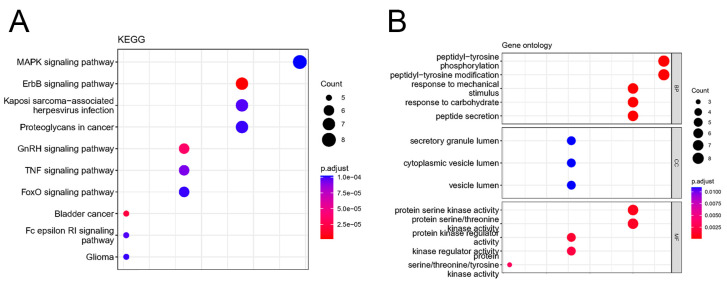
Functional enrichment analysis of DEOSGs. (**A**) KEGG annotations of DEOSGs (**B**) GO annotations of DEOSGs. BP, biological process; CC, cellular component; MF; molecular function.

**Figure 4 ijms-25-08353-f004:**
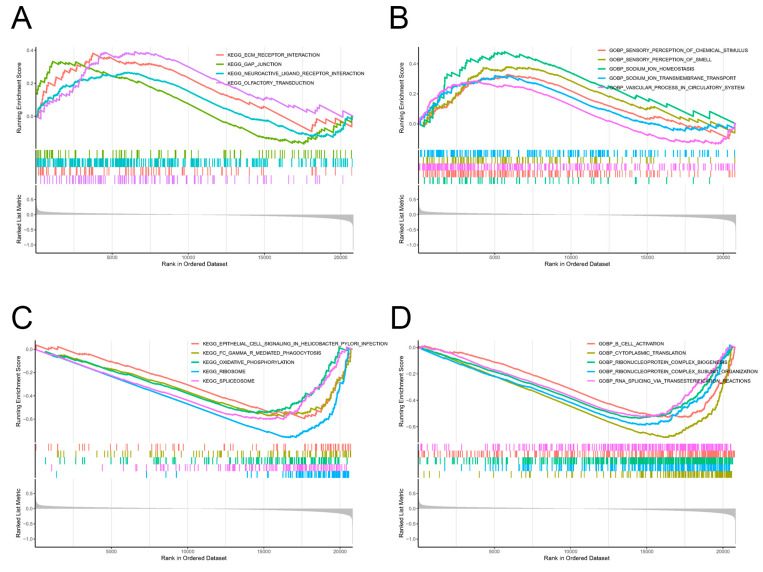
Gene set-enrichment analysis. Up-regulated genes enriched in (**A**) KEGG pathways and (**B**) GOBP. Down-regulated genes enriched in (**C**) KEGG pathways and (**D**) GOBP. GOBP, gene ontology biological process.

**Figure 5 ijms-25-08353-f005:**
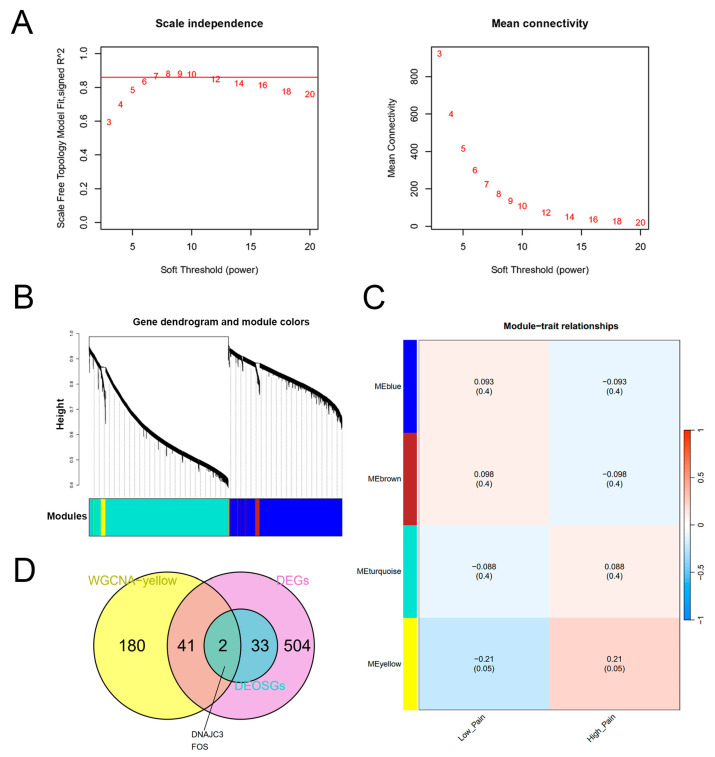
Weighted gene co-expression network analysis. (**A**) Scale independence and mean connectivity. (**B**) Gene dendrogram and modules in different colors. (**C**) Module-trait correlation analysis heatmap. (**D**) Venn diagram displaying key genes.

**Figure 6 ijms-25-08353-f006:**
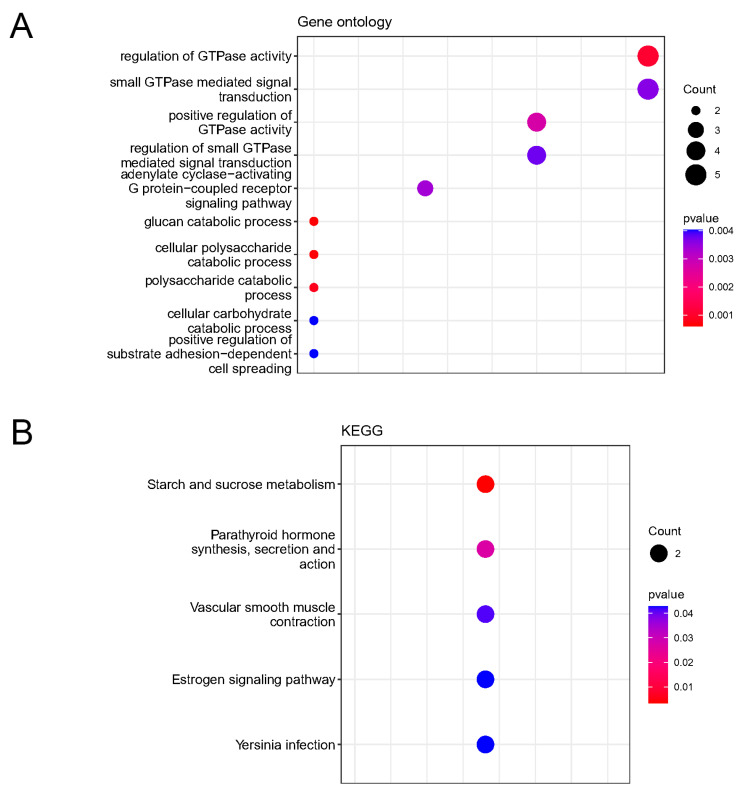
Functional enrichment of key genes. (**A**) GO annotation of key genes. (**B**) KEGG annotation of key genes.

**Figure 7 ijms-25-08353-f007:**
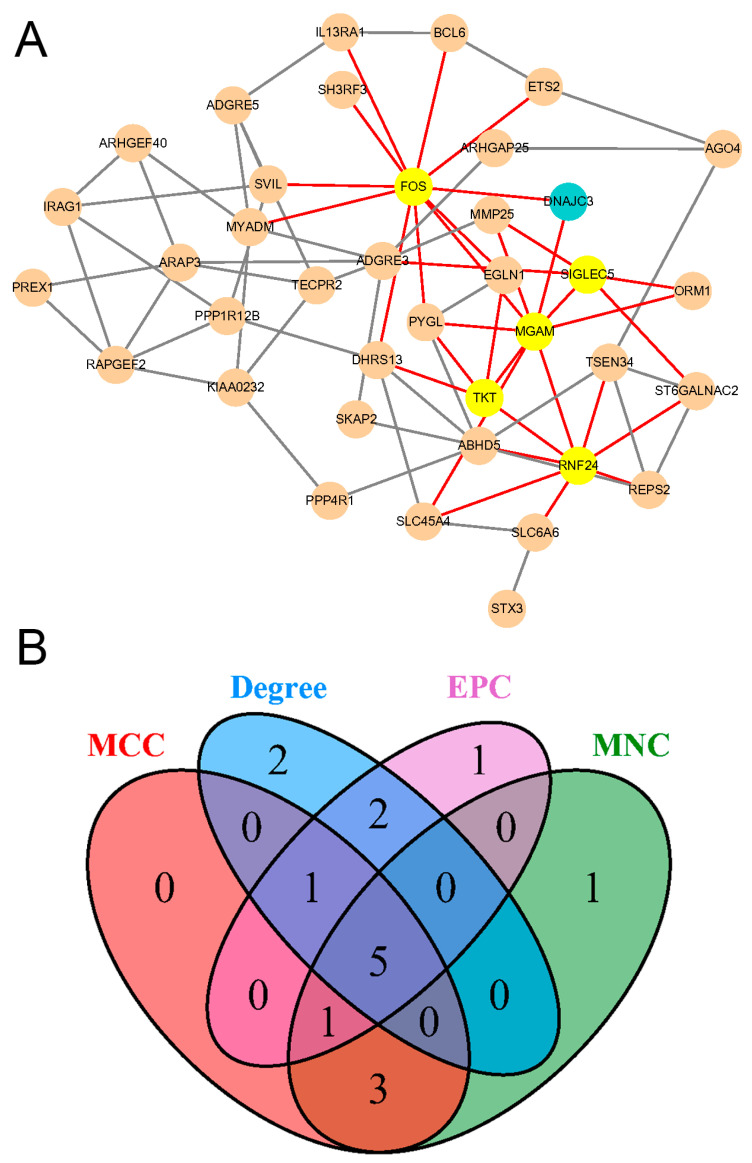
Protein–protein interaction network and hub genes. (**A**) The PPI network constructed with key genes (38 nodes and 78 edges). (**B**) Hub genes screened out by topological analysis algorithms. MCC, maximal clique centrality; EPC, edge percolated component; MNC, maximum neighborhood component.

**Figure 8 ijms-25-08353-f008:**
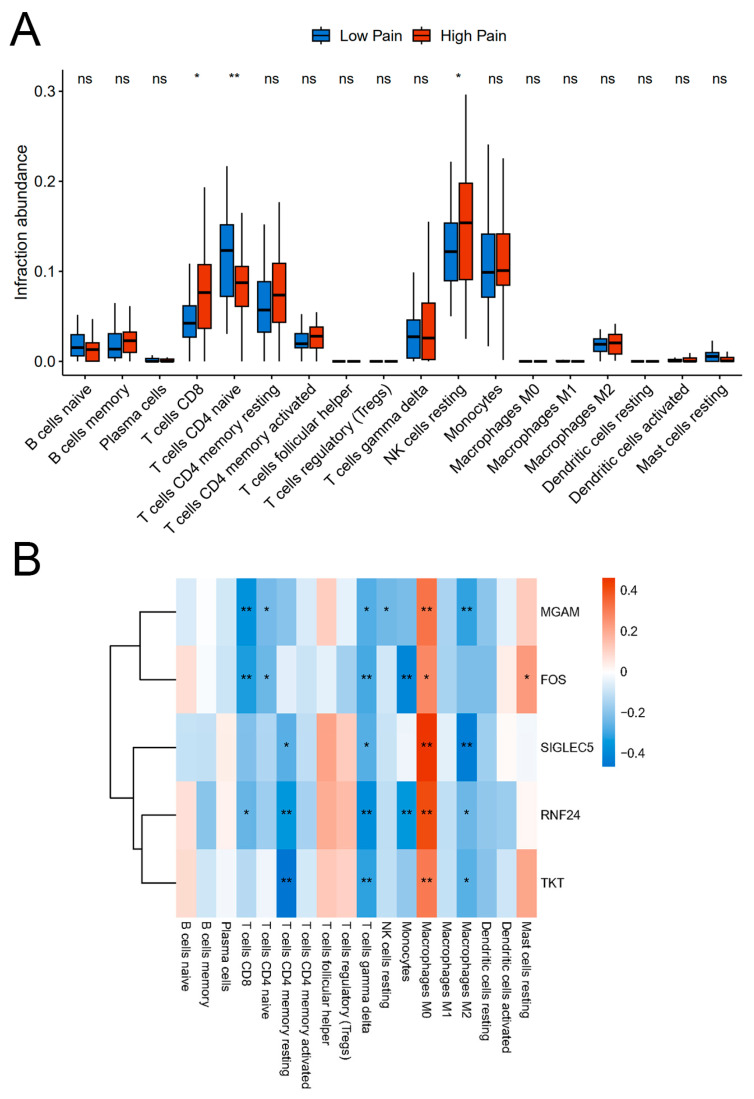
Immune infiltration analysis. (**A**) Differentially infiltrating immune cells between groups. (**B**) Correlations between differentially infiltrating immune cells and hub genes. * *p* < 0.05 and ** *p* < 0.01.

**Figure 9 ijms-25-08353-f009:**
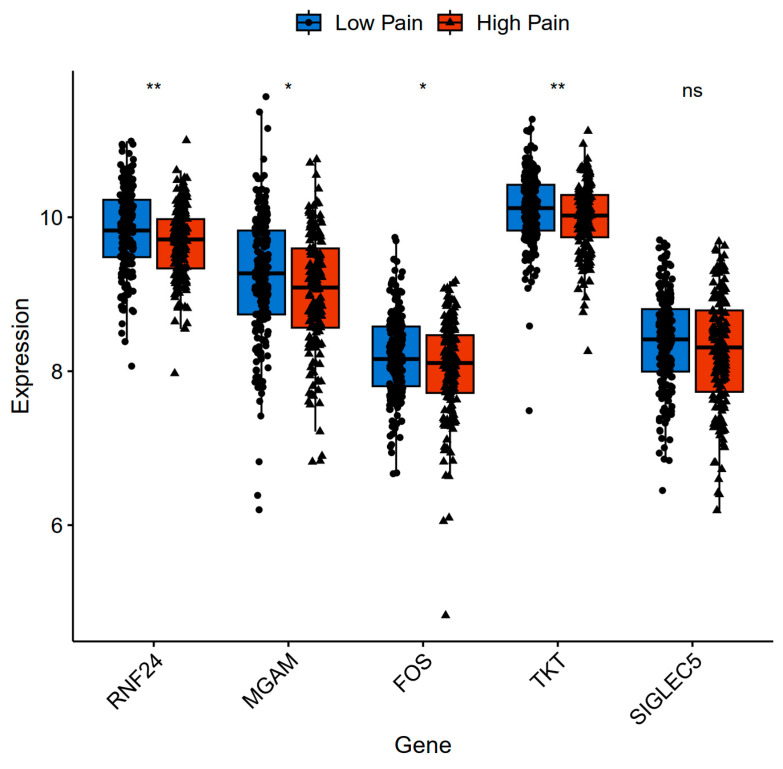
Gene expression level of hub genes in other gene set. * *p* < 0.05, ** *p* < 0.01, and ns non-significant.

**Figure 10 ijms-25-08353-f010:**
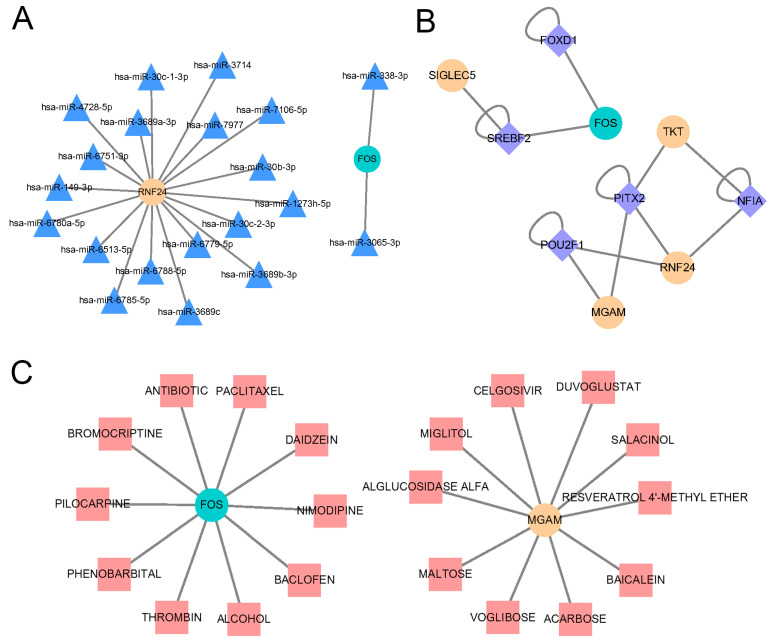
Regulatory networks for hub genes. (**A**) The microRNA–hub gene interaction network. (**B**) The transcription factor–hub gene interaction network. (**C**) The drug–hub gene interaction network. Yellow or green circles indicate hub genes.

## Data Availability

The original contributions presented in this study are included in this article/Appendix A, further inquiries can be directed to the corresponding author.

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
