# Peer review of "Identification of Oxidative Stress-Related Biomarkers for Pain–Depression Comorbidity Based on Bioinformatics"

_ijms, 2024, doi:10.3390/ijms25158353_

Round 1
Reviewer 1 Report (Previous Reviewer 1)
Comments and Suggestions for Authors
Reviewers’ Report:
Title: Identification of oxidative stress-related biomarkers for pain-depression comorbidity based on bioinformatics.
The manuscript from Zhang and co-workers describes the bioinformatics analysis of pain- depression comorbidity related samples from GSE127208 GEO dataset. It reports 43 key genes and 5 hub genes are differentially expressed between Low Pain vs High Pain groups in major depressive disorder (MDD) patients. Although the use of human samples is commendable, the study is primarily descriptive. The reported mechanistic insights into the pathology of MDD are rather limited. Overall, the presented dataset might be relevant for future research.
Major points:
The authors have resubmitted the manuscript with the same results in the first submission (ijms-3060769). Now they have provided a reference (19) that says that “No fold change cutoff was applied in analyses of brain samples because of the relatively subtle expression differences observed.” If this is the case, then the manuscript can be accepted now. As I have already provided the comments in previous revisions, so I don’t have extra comments now.
I recommend the manuscript to be accepted for publication in IJMS.
Author Response
Thank you for your valuable feedback on our manuscript.
Reviewer 2 Report (New Reviewer)
Comments and Suggestions for Authors
The authors have analysed previously published gene expression data, DEGs and specific genes related to oxidative stress in this study. Moreover, they found 43 key genes and 5 hub genes using different databases and techniques such as weighted gene networks, pathway and graph analysis. The authors showed that some of these genes might be valuable biomarkers for pain-depression comorbidity. Generally, the approach of analysis of this data is quite standard and does not go too far beyond even ready available automatic pipelines for analysis of gene expression data. Still, it is important to do the analysis carefully, taking into account all possible data problems. The authors did this quite well. Still it was not clear to me if the authors did data normalization at the very beginning.
Among critical comments I can say:
- I did not find in the paper discussion about the difference or similarity between they results they obtained and the results originally published by the authors of the GEO data, which were analyzed in this paper.
- The idea of analysis of finding hubs in the network, although generally accepted in the community, in my view has clear pitfalls. This way you would find genes that are very generic and involved in many processes which is usually not a very good biomarker, especially for such very special disease condition that is studied in this paper. And logically, the FOS gene, which was proposed here, is definitely not specific enough to be a good biomarker.
- Generally, the approach of taking simply all available tools and throwing them on the data, and then selecting genes that are simply found by all those tools has very little scientific merit. What is the biologically sound logic behind the tool selection and the whole approach?
Author Response
Please see the attachment.

This manuscript is a resubmission of an earlier submission. The following is a list of the peer review reports and author responses from that submission.
Round 1
Reviewer 1 Report
Comments and Suggestions for Authors
Reviewers’ Report:
Title: Identification of oxidative stress-related biomarkers for pain-depression comorbidity based on bioinformatics.
The manuscript from Zhang and co-workers describes the bioinformatics analysis of pain- depression comorbidity related samples from GSE110147 GEO dataset. It reports 43 key genes and 5 hub genes are differentially expressed between Low Pain vs High Pain groups in major depressive disorder (MDD) patients. Although the use of human samples is commendable, the study is primarily descriptive. The reported mechanistic insights into the pathology of MDD are rather limited. Overall, the presented dataset might be relevant for future research.
Comments and suggestions: There are few aspects that need to be addressed to further demonstrate the relevance of this study.
Major points:
1. Page 14, line 411: The authors have used GSE110147 dataset from GEO database. But, the GEO accession number belongs to different study rather than the current study. Please clarify. Please mention the date of data download.
2. No characterization of patient population (age, sex, MDD score etc.) is given. Please provide the demographic characteristics of the studied subjects included in the study in tabular form.
3. Figure 2A: How was the volcano plot prepared? It preferably made between log2(Fold Change) on X-axis and -log10(p-value) on Y-axis. What was the cutoff taken for log2 (Fold Change), please mark on the plot?
4. Figure 5C: The authors’ have selected yellow module with correlation value as 0.21 with p-value =0.05. The correlation value should be close to 1 to show high correlation.
5. Preparation of a supplementary table with the identified genes may improve the manuscript.
6. The validation is limited to dry lab data. Validation of the hub genes by another method such as RT-qPCR etc. can substantially improve the manuscript.
Minor points:
1. The introduction is short and quick, and I believe it doesn't adequately give a summary of the existing literature.
2. Figure 2A and 2C: The font size is not readable. Please enlarge the figures to get a better view of the results.
3. Figure 2B: The term ‘OSRGs’ is not elaborated in the text.
4. Figure 7: Please delete the extra text in the figure (GeneRatio, 004,003…)
5. Figure 8A: Please enlarge the figure and make the text visible.
6. The plagiarism rate is very high (42%) for the manuscript to accept. Please reduce it.
7. Please add some latest references.
8. The quality of the figures needs to be improved.
I recommend this manuscript for publication in IJMS after major revisions. The dataset taken for analysis is mismatching with the GEO database and the 580 DEGs used for downstream analysis was selected only based on p-values and not log2(Fold Change).
Reviewer 2 Report
Comments and Suggestions for Authors
The reviewed manuscript focused on the identification of oxidative stress-related biomarkers for pain depression based on gene expression analysis using modern bioinformatics techniques. The topic is interesting and actual. However, some shortcomings should be corrected before the manuscript is accepted. Below, I present my remarks.
1. The abstract should reflect the brief content of the research: Actuality, used methods, obtained results and their significance. To my mind, the current abstract does not correspond to this request. Please. rewrite it.
2. The Introduction section in your instance also contains the Literature review. In this case, it would be better to allocate the unsolved parts of the general problem and the main contribution of the authors' research at the end of this section.
3. The flowchart that is shown in Figure 1 should be presented in the section Material and Methods. Moreover, the manuscript will be more readable if this section, in this case, is given after the introduction, before the results section. The title of Figure 1 will look better if the "Flowchart of the study", but not this study.
4. The title of all figures should be given after the appropriate Figure. Please correct this.
Round 2
Reviewer 1 Report
Comments and Suggestions for Authors
Revision 2:
1. Figure 1: Please change the GEO accession number in the figure also.
2. Table S1: The abbreviation used in diagnosis column of the table needs to be defined as footnotes.
3. Figure 2A: The cutoff for log2 (Fold Change) is still not marked on the figure. The figure is not acceptable only with p-value. The X-axis title is still the same. It should be changed to log2 (Fold Change). The reference provided by the authors’ for “no fold change cutoff” is Sieriebriennikov et al., 2020 (Ref. 19). The data provided in Table 2 of this manuscript has taken a minimum log Fold Change of 0.7 and non-significant (NS) if value if below 0.7. So, selection of DEGs without fold change cutoff is not justified.
4. Figure 5C: The correlation value of 0.2 is not strong, its weak correlation. So, selecting module yellow (MEyellow) is not justified. The module cannot be selected if it is best but it should have higher correlation value also preferably close to 1.
5. Please mention the plagiarism rate after first revision.
I recommend a second major revision of the manuscript.
Round 3
Reviewer 1 Report
Comments and Suggestions for Authors
Revision 3:
The revision provided by the authors’ is still not convincing. The log2 (Fold Change) cutoff chosen is 0.08, which is not permissible. The cutoff should be between 0.5 to 1 after which all the downstream results will change. The chosen cutoff should be validated using independent methods such as qRT-PCR.
I should reject the manuscript as it is not able to provide suitable responses and missing wet lab validation of the results.